# Asymmetric Synthesis of Tetrasubstituted α-Aminophosphonic Acid Derivatives

**DOI:** 10.3390/molecules26113202

**Published:** 2021-05-27

**Authors:** Aitor Maestro, Xabier del Corte, Adrián López-Francés, Edorta Martínez de Marigorta, Francisco Palacios, Javier Vicario

**Affiliations:** 1Departamento de Química Orgánica I, Centro de Investigación y Estudios Avanzados “Lucio Lascaray”-Facultad de Farmacia, University of the Basque Country, UPV/EHU Paseo de la Universidad 7, 01006 Vitoria-Gasteiz, Spain; aitor.maestro@ehu.eus (A.M.); xabier.delcorte@ehu.eus (X.d.C.); adrian.lopez@ehu.eus (A.L.-F.); edorta.martinezdemarigorta@ehu.eus (E.M.d.M.); 2Stratingh Institute for Chemistry, University of Groningen, 9747 AG Groningen, The Netherlands

**Keywords:** asymmetric synthesis, α-aminophosphonic acid, tetrasubstituted carbons, diastereoselective, enantioselective, α-aminophosphonates

## Abstract

Due to their structural similarity with natural α-amino acids, α-aminophosphonic acid derivatives are known biologically active molecules. In view of the relevance of tetrasubstituted carbons in nature and medicine and the strong dependence of the biological activity of chiral molecules into their absolute configuration, the synthesis of α-aminophosphonates bearing tetrasubstituted carbons in an asymmetric fashion has grown in interest in the past few decades. In the following lines, the existing literatures for the synthesis of optically active tetrasubstituted α-aminophosphonates are summarized, comprising diastereoselective and enantioselective approaches.

## 1. Introduction

α-Amino acids are a key structure in living organisms as the essential part of proteins and peptides. Many α-amino acid derivatives are used in daily life, such as the sweetener aspartame, penicillin-derived antibiotics or antihypertensive enalapril. Due to the relevance of α-amino acids in nature, a vast number of methods for the synthesis of natural and non-natural α-amino acids have been developed [1,2,3]. Within the most relevant α-amino acid mimetics, α-aminophosphonic acids are the result of a bioisosteric substitution of the planar carboxylic acid by a phosphonic acid group in α-amino acid structures (Figure 1).

This isosteric replacement is of great interest since, due to the tetrahedral configuration of the phosphorus atom, α-aminophosphonic acid derivatives can behave as stable analogues of the transition state for the cleavage of peptides, thus inhibiting enzymes involved in proteolysis processes and, consequently, they display assorted biological activities [4,5,6,7]. In particular, a number of α-aminophosphonic acid derivatives have found applications as agrochemicals [8,9], as well as antimicrobial [10,11,12], antioxidant [13,14,15] or anticancer agents (Figure 2) [16,17,18].

The thalidomide disaster was a shocking revelation of the strong dependence of the biological activity of chiral substrates into their absolute configuration. This dependence is also evident for α-aminophosphonic acid derivatives and, as examples, (*R*)-phospholeucine exhibits a stronger activity as leucine-peptidase inhibitor than its enantiomer (Figure 3) [19,20], and the phosphopeptide (*S*),(*R*)-alaphosphalin has a more efficient antibiotic activity than its other three possible isomers [21,22].

α-Aminophosphonic acids are usually obtained from the hydrolysis of their phosphonate esters and, for this reason, the development of efficient synthetic methodologies to access enantioenriched α-aminophosphonates has become an imperative task in organic chemistry. The existing literature to date in this field is mostly related to the synthesis of trisubstituted α-aminophosphonates and the examples illustrating asymmetric strategies leading to the formation of the tetrasubstituted substrates are scarce [23,24]. The efficient formation of quaternary centers is known as a critical challenge in organic synthesis [25,26,27] and the formation of tetrasubstituted centers from ketimines was for a long time unachievable. The poor electrophilic character of the ketimine group and the additional steric hindrance on the substrate that results in a decreased reactivity are the two main challenges to overcome. In addition, the enantiotopic faces of ketimine substrates are not as easily discriminated as those of aldimines if asymmetric syntheses are required [28].

The existing methods for the synthesis of tetrasubstituted α-aminophosphonates can be classified in three main groups, depending on the type of bond created in the key reaction leading to their formation Figure 4). The first of those approaches implies the use of strategies that entail C-C bond formation, either through the addition of carbon nucleophiles to α-phosphorylated imines (Figure 4a) or functionalization of α-aminophosphonate anions with electrophiles (Figure 4b). In addition, the most straightforward method for the synthesis of α-aminophosphonates comprises reactions that imply C-P bond formation through the addition of phosphorus nucleophiles to ketimines (Figure 4c). Another alternative to these routes consists of processes that involve C-N bond formation, which are carried out mainly through electrophilic amination reactions (Figure 4d).

In the following lines of this review, the existing methodologies regarding the asymmetric synthesis of tetrasubstituted α-aminophosphonates are summarized. The synthetic routes for the preparation of these compounds are classified into diastereoselective and enantioselective methodologies and grouped by the type of bond formed in the key step.

## 2. Diastereoselective Synthesis of Tetrasubstituted α-Aminophosphonic Acid Derivatives

### 2.1. C-C Bond Formation

A simple strategy for the preparation of tetrasubstituted α-aminophosphonates is the functionalization of tetrasubstituted α-aminophosphonates, taking advantage of the acidic nature of the hydrogen atom adjacent to the phosphorus substituent. Using this approach, Seebach described in 1995 the first example of a stereoselective synthesis of tetrasubstituted α-aminophosphonic acids [29]. As shown in Scheme 1, racemic imidazolidinone **2** is first obtained starting from glycine ester **1**, by formation of the amide derivative with dimethylamine and subsequent condensation of the amino group with pivalaldehyde. Then, a kinetic resolution using (*R*)-mandelic acid **3** is carried out, allowing the isolation of diastereomeric salt (*R*, *R*)-**4**, which is treated first with NaOH and then with Boc_2_O and DMAP to yield enantiomerically pure imidazolidinone **5** [30].

Then bromide **6** is formed via radical halogenation, and the subsequent Arbuzov reaction with trimethylphosphite leads to a single isomer of tertiary α-aminophosphonate **7** in moderate yield [31]. Next, in the key step, compound **7** is treated with a strong base and an alkyl, allyl or benzyl halide, leading to the formation of tetrasubstituted α-aminophosphonates **8** in good yields and excellent diastereoselectivities (72–83%, >98:2 dr). In all cases, the electrophile reagent approaches from the less hindered face of imidazolidinone ring, in an *anti*-addition. In order to obtain the acyclic α-aminophosphonic acid derivative **9**, the authors performed a reduction of the amide carbonyl in **8** and a hydrolysis of the resulting intermediate in aqueous HCl [29].

In addition, the authors extended their methodology to the use of imidazolidines, instead of amides **7**, obtaining, after the addition of benzyl, methyl or allyl halides, *cis* tetrasubstituted α-aminophosphonates in moderate yields and diastereoselectivities (33–53%, 1.3:1–3.8:1 dr). However, the use of ethyl, propyl and butyl halides gave *trans* products in better yields and diastereomeric ratios (45–60%, 1:17 ≤ 1:50 dr).

A similar procedure was developed by Davis for the synthesis of pyrrolidine-derived α-aminophosphonates **17** (Scheme 2). In this case, sulfinyl imine **10** is attacked by the enolate derived from ethyl acetate, to obtain *N*-sulfinyl β-amino ester **11** in a diastereoselective fashion. Then, the addition of lithium methyl phosphonate leads to *N*-sulfinyl δ-amino β-ketophosphonate **12** in very good yield. Next, the sulfinyl protecting group is easily removed and replaced with a Boc group, obtaining amide **13** which, after treatment with 4-acetamidobenzenesulfonyl azide, provides the corresponding α-diazo derivative **14**. Finally, the treatment of α-diazophosphonate **14** in the presence of Rh_2_(OAc)_4_ catalyst yields *cis* pyrrolidine phosphonate **15**, as the major diastereoisomer (68%, 81:19 dr) [32]. Tertiary α-aminophosphonate **15** can be functionalized, with retention of the configuration, using a strong base and allyl bromide, providing tetrasubstituted α-aminophosphonate **16**. Although substrate **16** is obtained as a mixture of rotamers, the removal of *N*-Boc group renders pyrrolidin-3-one **17** as a single isomer. Additionally, acyclic α-amino α-ketophosphonate **18** can be also prepared after a ring-opening process via a Pd-catalyzed hydrogenation [33].

Following the same principle, Amedjkouh described the synthesis of bicyclic α-aminophosphonate **24** (Scheme 3). In this case, the synthetic methodology starts with the preparation of oxazolopyrrolidine phosphonate **23** from (*R*)-phenylglycinol (**19**), benzotriazole (**20**), and 2,5-dimethoxytetrahydrofuran (**21**), obtaining enantiomerically pure oxazolopyrrolidine **22**, by formation of succinaldehyde from the hydrolysis of furan derivative **21** and subsequent multicomponent reaction with substrates **19** and **20**. Then, tertiary oxazolopyrrolidine phosphonate **23** is formed through Arbuzov reaction of oxazolopyrrolidine **22** and triethyl phosphite [34,35]. The treatment of α-aminophosphonate **23** with butyllithium followed by the addition of alkyl halides results in the formation of tetrasubstituted α-aminophosphonates **24** with total retention of the configuration, and yields that are moderate to good, when using aliphatic halides (35–81%), but low if benzyl halide is used (10%). Finally, the elimination of the chiral auxiliary via catalytic hydrogenation affords optically pure phosphoproline derivative **25** [35].

According to the authors, the high diastereoselectivity observed for this transformation is related to a proposed transition state **TS1** (Figure 5). Thus, the lithium ion is coordinated with the phosphonate oxygen and the tertiary nitrogen atoms, forming a five-membered ring pseudocycle, where the σ* P=O acceptor orbital lies in parallel to the lone pair of the anion, which is therefore stabilized by hyperconjugation. Under this conformation, the functionalization with the alkyl group occurs with retention of configuration.

Another possibility for the asymmetric formation of tetrasubstituted α-aminophosphonates implying C-C bond formation relies on the addition of carbon nucleophiles to chiral ketimines. In this context, our research group described in 2013 a methodology for the preparation of tetrasubstituted α-aminophosphonates through the addition of carbon nucleophiles to α-phosphorated ketimines **29** (Scheme 4) [36]. First, Pudovic reaction of TADDOL-derived chiral phosphite **26** with imine **27** affords trisubstituted α-aminophosphonate **28** as a mixture of diastereoisomers (93%, 1:1 dr). α-Aminophosphonate **28** is then transformed into the enantiopure chiral ketimine **29** through a formal oxidation, consisting of an initial chlorination, followed by a β-elimination of hydrogen chloride using a polymeric base. Then, organometallic species react fast with α-iminophosphonate **29**, delivering tetrasubstituted α-aminophosphonates **30** in very good yields. Better results in terms of diastereoselectivity are obtained in this reaction when aliphatic Grignard reagents (R = Me, 80%, 94:6 dr) are used if compared to the aromatic substrate R = 2-naphthyl, 81%, 55:45 dr). Additionally, the hydrolysis of the tosyl and chiral auxiliary group affords tetrasubstituted α-aminophosphonic acid (*S*)-**31**.

In the proposed transition state, depicted in Figure 6, the phosphorus-containing seven-membered ring adopts a more stable boat conformation, which is fixed by the *trans* configuration of the five-membered fused ring. Under this conformation, the two heteroatoms must embrace the more stable equatorial orientation, forcing the two hydrogen atoms to the axial positions. According to this model, the nucleophilic attack to the *Re*-face is substantially favored, due to the presence of the axial phenyl groups blocking the *Si* face.

Menthol-derived phosphonate imines **32** are another kind of useful chiral electrophiles used in the diastereoselective preparation of tetrasubstituted α-aminophosphonates (Scheme 5). For example, the use of proline (**I)** as a chiral catalyst in the addition of acetone (**33**) gives α-aminophosphonate **34** in good yield and diastereoselectivity (86%, 1:30 dr). The authors also describe the addition of other nucleophiles to ketimines **32**, such as pyrroles **35**, indole (**37**), and nitromethane, to yield α-aminophosphonate derivatives **36**, **38** and **39**, although in these cases, they observe lower diastereoselectivities [37].

The same research group used also chiral *N*-methylbenzylimines **41** in cycloaddition reactions (Scheme 6) [38] for the asymmetric preparation of tetrasubstituted α-aminophosphonates **43**. In his case, the synthesis of chiral imine **41** comprises an initial treatment of (*S*)-1-phenylethylamine ((*S*)-**40**) with trifluoroacetic acid and triphenylphosphine, in the presence of trimethylamine in order to form haloimine **41**. Then, the Arbuzov reaction with triethyl phosphite, yields the corresponding fluorinated α-ketiminophosphonate **42**. Upon treatment with diazomethane, ketimine **42** undergoes a cycloaddition reaction that leads the formation of triazoline-derived α-aminophosphonate **43** in good yield but with moderate diastereoselectivity (83%, 2.5:1 dr). Although the authors did not assign the configuration of the major isomer, both can be separated through chromatography.

### 2.2. C-P Bond Formation

Another common strategy for the preparation of tetrasubstituted α-iminophosphonates comprises the addition of phosphorus nucleophiles to ketimines. For example, Davis described in 2001 the use of *p*-toluensulfinyl imines **44** as starting materials for the diastereoselective preparation of tetrasubstituted α-aminophosphonates **45** (Scheme 7) [39]. In this report, chiral ketimines **44** are treated with lithium diethyl phosphite at low temperature to obtain α-aminophosphonates **45** in excellent yields and diastereoselectivities (71–97%, 82:18–99:1 dr). In the last step, enantiomerically pure α-aminophosphonic acids **46** can be isolated by simple hydrolysis using hydrochloric acid.

The high degree of diastereoselectivity in this reaction is explained by the authors as a transition state where there is a chelation of the lithium cation to both, the sulfinyl and phosphite oxygen atoms, in a seven-membered twisted-chair transition state. As shown in Figure 7, **TS3** is favored because, under this conformation, the bulky aryl group adopts an energetically favored equatorial position if compared to **TS4**, where the aromatic substituent group must assume an energetically less favorable axial position.

Following a similar methodology, the same research group describes also the preparation of tetrasubstituted phosphoproline derivative **50** (Scheme 8) [40]. In this case, the treatment of oxo-sulfinimine **47** with lithium diethyl phosphite gives α-aminophosphonate **48**, which after treatment with hydrochloric acid, forms cyclic tetrasubstituted α-aminophosphonate **49** in good yield. Then, a *syn* addition of molecular hydrogen from the less hindered face yields phosphoproline derivative **50** with 50% enantiomeric excess.

Based on the same principle, Yuan’s group used *tert*-butylsulfinyl imines **51** as chiral auxiliaries for the synthesis of tetrasubstituted α-aminophosphonates **52**, via nucleophilic addition of phosphites (Scheme 9) [41,42]. Substrates **52** are obtained with high yields and diastereoselectivities in all cases (70–85%, 86:14 ≥ 98:2 dr), using different alkyl phosphites (R^2^ = Me, Et, *n*-Pr) and several alkyl or aromatic substituents in sulfinimine **51**. In an identical way as described by Davis, substrates **52** can be transformed into α-aminophosphonic acids **53** by a simple hydrolysis. For this transformation, the authors propose the plausible transition state **TS5**, where the potassium cation is chelated to the sulfinyl and phosphonate oxygens, and the nucleophilic attack of the phosphite nucleophile occurs from the less hindered face, opposite to the *tert*-butyl group.

A few years later, Ellman’s research group described a modification of this reaction, using potassium bis(trimethylsilyl)amide (KHMDS), which favors the solubility obtaining in this way α-aminophosphonates **52** with better reaction conversions and in excellent yields but lower diastereoselectivities (88–95%, 12:1–99:1 dr) [43].

In addition, Yuan’s group used this strategy, employing chloro-substituted sulfinyl imines **54** for the preparation of three-, four- and five-membered cyclic tetrasubstituted α-aminophosphonates **56** (Scheme 10) [42]. The reaction proceeds efficiently using different imines **54** (R = Me, Ph; n = 1, 2, 3) and heterocyclic substrates **56** can be obtained through intermediate **55** in good yields and diastereoselectivities (75–83%, 71:29–92:8 dr). In this transformation, a strong dependence of the diasteroselectivity on the size of the cycle is observed, obtaining an excellent dr (92:8) for a three-membered cyclic substrate, while a drop into the diastereoselectivity is observed for the four-membered derivative (89:11 dr) and even lower dr values are obtained for five-membered heterocycles (71:29 dr).

Likewise, in 2014, Liu and colleagues extended this strategy to the nucleophilic addition of diphenyl phosphite to fluorine-substituted α,β-unsaturated sulfinimines **57**, in this case in the presence of a rubidium catalyst (Scheme 11) [44]. Allyl α-aminophosphonates **58** are obtained in good yields and diastereoselectivities (56–87%, 75:25–92:8 dr) with different fluoroalkyl substituents. In addition, the selective deprotection of the sulfinyl group in acidic media, to produce α-aminophosphonate **59** in good yield, is described.

In 2017, Cramer reported an additional example of an asymmetric addition of phosphorus nucleophiles to imines for the preparation of tetrasubstituted α-aminophosphonates (Scheme 12) [45]. In this work, first chiral imine **61** is obtained in good yield and enantiomeric excess (90%, 97% ee) from imidoyl chloride **60** in the presence of a palladium, catalyst **II** and CsOAc. Then, via a boron trifluoride-mediated hydrophosphonylation reaction of imine **61**, α-aminophosphonate **62** is formed in good yield as a single diastereoisomer.

A related strategy for the asymmetric induction in the preparation of tetrasubstituted α-aminophosphonates consists of the use of acetal-derived iminium salts as electrophiles. In 2000, Fadel and colleagues detailed a one-pot synthesis of cyclopropane α-aminophosphonates **65** (R^1^ = Me) using this methodology (Scheme 13) [46]. In this example, the sequence starts with the cyclization of bromoester **63** (R^1^ = Me) in the presence of sodium and TMSCl, to obtain silylated acetal **64**. Then, deprotected hemiacetal intermediate **66** is formed by an alcoholysis in presence of a catalytic amount of an acid source (TMSCl or AcOH), followed by the reaction with (*S*)-1-phenylethylamine ((S)-**40**), to furnish α-amino alcohols **67**. Under acidic conditions, intermediate **67** is converted into iminium species **68**, which undergo a nucleophilic addition of phosphite (R^2^ = Me, Et) from the less hindered face of the C=N bond, to provide finally diastereoisomeric phosphonates **65** (R^1^ = Me) in good yields and diastereoselectivities (60–82%, 80:20–88:12 dr).

A few years later, the authors extended the scope of this reaction to differently substituted acetals **64** (R^1^ = Et, Bn, *^i^*Pr, *^t^*Bu), obtaining cyclopropane-derived α-aminophosphonates **65** in good yields and excellent diastereoselectivities (56–78%, 76:24–100:1 dr). Remarkably, the use of a *tert*-butyl substituent provided a single diastereoisomer (100:1 dr) [47].

Following the same approach, Faldel’s group described later the synthesis of spirocyclic α-aminophosphonates **70** (Scheme 14) [48]. Starting also from silylated acetal **64**, deprotected acetal intermediate **66** is again formed by an alcoholysis and then, iminium species **72** is obtained by reaction with (*R*)-phenylglycinol under acidic conditions. The subsequent nucleophilic addition of triethyl phosphite gives spirophosphonate **69**, by means of an intramolecular transesterification, with good yield and diastereoselectivities (71%, 89:11 dr). However, the use of norephedrine as a chiral source results in a drop in both yield and diastereomeric ratio (36%, 78:22 dr). It must be pointed out that substrates **69** are obtained as a mixture of epimers (80:20), due to the presence of an additional chiral center at the phosphorus atom. The major diastereoisomer can be isolated and further transformed into enantiopure cyclopropane-derived aminophosphonic acid **70**, by an initial hydrogenolysis reaction, followed by hydrolysis of phosphonate group.

In 2007, Fadel described also a similar process, in this case using heterocyclic iminium salts (Scheme 15) [49]. In this approach, *N*-Boc-protected piperidinone **73** is treated with (*S*)-1-phenylethylamine ((*S*)-**40**) in presence of acetic acid, followed by the nucleophilic addition of triethyl phosphite, leading to the formation of tetrasubstituted α-aminophosphonates **74** and **75** as an inseparable mixture of diastereoisomers (75%, 60:40 dr).

Then, after the hydrolysis of the *N*-Boc protecting group, diastereoisomers **76** and **77** are formed, which in this case can be separated. In addition, the hydrogenolysis and hydrolysis reactions of each diastereoisomer gives enantiopure α-aminophosphonic acids **78** and **79**.

Along the same line, the same research group described the preparation of bicyclic tetrasubstituted α-aminophosphonates **84** starting from ketone acetals **80** (Scheme 16) [50]. The esterification reaction of acetals **80** with (*S*)-phenylalanine derivative **81**, which acts as chiral auxiliary, leads to intermediate **82**, which is cyclized to form imine substrates **83** as a mixture of diastereoisomers. The formation of iminium cation in the presence of triethyl phosphite gives bicyclic tetrasubstituted α-aminophosphonates **84** in good yields and excellent diastereoselectivities (46–77%, 89:11–99:1 dr). These α-aminophosphonates are transformed into cyclic serine analogues **87** through their oxidation, and further hydrolysis of the intermediate imine **85**/enamine **86** mixture.

In this case, the authors justify the high level of diastereoselectivity by an equilibrium between two iminium epimers in **TS6** and **TS7** through the parent enamine salt (Figure 8). It is estimated that the chair–boat conformation (**TS6**) is favored in 2.14–5.56 kcal/mol relative to the epimeric twist boat–boat conformation (**TS7**), resulting in the kinetic addition of the phosphite to the less hindered *Si*-face of the iminium species in **TS6**.

In addition, the use of bicyclic iminium salts has been reported for the asymmetric preparation of cyclic α-aminophosphonates **90** (Scheme 17). Chiral cyclic imines **89** are synthesized from diamine **88** and ketoesters and their subsequent treatment in toluene with dialkyl phosphites gives tetrasubstituted α-aminophosphonates **90** in high yields and diastereoselectivities (78–95%, 68:32–98:2 dr) **[51,52]**. However, if imines **89** are activated with bromotrimethylsilane, they are supposed to form an iminium ion, which is reactive towards tris(trimethylsilyl) phosphite and then, α-aminophosphonic acid derivatives **91** can be obtained in high yields and diastereoselectivities (70–99%, 85:15–98:2 dr) [51].

Other useful strategy for the diastereoselective synthesis of tetrasubstituted α-aminophosphonates with C-P bond formation, complementary to the hydrophosphonylation of chiral imines, is the addition of chiral phosphorus nucleophiles to activated ketimines. For example, in 2011, Chen and Miao used a multicomponent Kabachnik–Fields reaction of phosphorylated chiral nucleophile **92**, diethyl phosphoramidate **93** and ketone **94** to obtain α-aminophosphonate **95** in a diastereoselective fashion (Scheme 18) [53]. The authors propose that the nucleophilic addition in **TS9** is expected to be less favored, compared to the addition proposed in **TS8**, where the chiral dioxaphospholanedicarboxylate **92**, which plays a crucial role in the control of the diastereoselectivity, reacts from the sterically less hindered face.

A different methodology for the asymmetric formation of tetrasubstituted α-aminophosphonates that involves C-P bond formation was reported by Hammershmidt, where a phosphoramidate-α-aminophosphonate rearrangement is described, leading to the formation of diverse α-aminophosphonates **98** in moderate to good yields and excellent stereocontrol (38–80%, 96–99% ee) (Scheme 19) [54,55]. This route involves *N*-Boc protection of phosphoramides **96**, and metalation with *sec*-butyllithium to form the corresponding carbanion **99**. The rearrangement of the phosphorous substituent and the final quenching with acetic acid provides tetrasubstituted α-aminophosphonates **98**.

### 2.3. C-N Bond Formation

The introduction of nitrogen reagents into the skeleton of phosphonates is also an alternative methodology that can be useful for the preparation of tetrasubstituted α-aminophosphonates. In this regard, in 1999 Davis and colleagues applied successfully the aza-Darzens reaction for this purpose (Scheme 20) [56]. Starting from chiral sulfinyl imine **101** and diethyl 1-chloroethylphosphonate (**102**), initially, a mixture of three isomers of α-chloro-β-amino adducts **103**–**105** is obtained.

The major isomer **103** can be isolated after chromatography and then, in the presence of sodium hydride, enantiomerically pure aziridine **106** is obtained. After the elimination of the chiral auxiliary group with TFA, followed by ring-opening via hydrogenolysis, enantiopure tetrasubstitued α-aminophosphonate (*R*)-**107** is obtained. For the preparation of the opposite enantiomer, the side mixture of α-chloro-β-amino adducts **104** and **105** is used. After the hydrolysis of the chiral auxiliary group and the subsequent ring-opening of the corresponding aziridine intermediate, tetrasubstitued α-aminophosphonate (*S*)-**107** is obtained.

Continuing with strategies that entail C-N bond formation, Curtius rearrangement is also a useful method for the introduction of amino groups starting from carboxylic acids, which can be used for the preparation of tetrasubstitued α-aminophosphonates. For example, in 1999, Le Corre used enantiomerically pure chiral sulfate **108** and phosphorylated malonate derivative **109** for the preparation of cyclopropane phosphonate **110** (Scheme 21) [57].

Then, the ester group is hydrolyzed, to obtain carboxylic acid substituted phosphonate **111** which, after activation of the acid with thionyl chloride and the addition of sodium azide, leads to acyl azide species **112**/**113**. At this point, Curtius rearrangement gives isocyanate **114**, which is captured by means of the in situ addition of benzyl alcohol, to afford *N*-protected amino ester **115**. Finally, the benzyl protecting group is hydrolyzed yielding cyclopropane-derived tetrasubstituted α-aminophosphonate **116**.

Ito’s group also used Curtius rearrangement for the preparation of tetrasubstitued α-aminophosphonate **123** (Scheme 22) [58]. The rhodium-catalyzed conjugate addition of cyanophosphonate **117** to acrolein (**118**) leads to the formation of aldehyde **119** with high yield and enantiomeric excess (80%, 92% ee). Compound **119** is then treated with phosphonium ylide **120**, and the newly formed C=C bond, prepared through the Wittig olefination, is directly hydrogenated to obtain cyanophosphoate **121**. The acidic hydrolysis of the nitrile moiety in this substrate followed by an in situ esterification of the carboxylic acid intermediate with diazomethane affords phophorated methyl ester **122**. Finally, the methoxycarbonyl group in **122** is selectively hydrolyzed under basic conditions, and the resulting carboxylate treated with diphenyl phosphoroazidate, which by means of a Curtius rearrangement followed by trapping with benzyl alcohol, affords tetrasubstitued α-aminophosphonate **123** (81%, 88% ee).

Following a similar approach, a few years later, Krawczyk and colleagues reported an analogous reaction, in which the synthetic route starts with the reaction between cyclic sulfate **124** and ethyl diethoxyphosphorylacetate **125**, to afford phosphorated ester **126** as a single diastereoisomer (Scheme 23) [59]. In order to perform the Curtius rearrangement, first the ester group needs to be hydrolyzed to form carboxylic acid **127**, and then the addition of diphenylphosphoryl azide (DPPA) affords isocyanate **128**, which is immediately captured as carbamate **129** by the in situ addition of ethanol. In addition, the benzyl and carbamateprotecting groups can be eliminated via hydrogenolysis and hydrolysis, respectively, affording enantiopure α-aminophosphonic acid derivative (1*R*,2*S*)-**130** in excellent yield.

For the preparation of the opposite enantiomer, (1*S*,2*R*)-**130,** the authors used a complementary strategy as depicted in Scheme 24. In this case, the synthesis of racemic lactone **132** is performed by treatment of epibromohydrin **131** with malonate derivative **125** in the presence of sodium hydride. Then, the reaction of lactone **132** with (*R*)-1-phenylethylamine **40** gives products (1*S*,2*R*,1′*R*)-**133** and (1*R*,2*S*,1′*R*)-**133** as a mixture of diastereoisomers, which can be separated by column chromatography. Once pure (1*S*,2*R*,1′*R*)-**133** is isolated, the hydrolysis of the amine in presence of sulfuric acid yields enantiomerically pure lactone (1*S*,5*R*)-**134,** which is then treated with saturated methanolic ammonia followed by an acylation reaction to obtain amide **135**. The lead tetraacetate-mediated Hoffmann rearrangement in *tert*-butyl alcohol gives carbamate **136**, which in presence of potassium carbonate yields *N*-Boc aminocyclopropane phosphonate **137**. Finally, the sequential treatment of **137** with TFA and bromotrimethylsilane affords α-aminophosphonic acid derivative (1*S*,2*R*)-**130** in excellent yield [59].

## 3. Enantioselective Synthesis of Tetrasubstituted α-Aminophosphonic Acid Derivatives

### 3.1. C-C Bond Formation

The first example of an enantioselective synthesis of tetrasubstituted chiral α-amino phosphonates was reported in 1999 by Ito’s group (Scheme 25) [60]. The reaction consists of an asymmetric palladium–**IV**-catalyzed allylation of racemic β-keto-α-aminophosphonates **138** that allows the obtaining of optically active α-amino phosphonates **140** in moderate to good yields and enantiocontrol (27–87%, 46–88% ee). In addition, the authors also report the subsequent diastereoselective reduction of the ketone moiety, affording β-hydroxy-α-amino phosphonates **141**. Remarkably, when the reaction is carried out in methanol at low temperature, using sodium or tetra *n*-butylammonium borohydrides as reducing reagents, the **141**-syn isomer is obtained (74–89%, 74:26 to 82:18 *syn*/*anti* ratio). In contrast, the reaction in *tert*-butyl alcohol at 50 °C, using sodium borohydride, affords the opposite isomer (78%, 15:85 *syn*/*anti* ratio).

Due to their strong nucleophilic character, the use of nitroalkane enolates has been widely extended in organic chemistry in the functionalization of aldehydes or imines since the Henry reaction was reported in 1895. Several examples regarding the enantioselective nucleophilic addition of α-nitrophosphonates to different electrophile reagents for the synthesis of α-aminophosphonates have been reported in recent years. In this context, the first example of this reaction was reported in 2008 by Johnston and colleagues (Scheme 26) [61]. The reaction consists of a Brönsted acid **V**-catalyzed addition of trisubstituted nitrophosphonates **142** to *N*-Boc aldimines **143**, which leads to chiral phosphonates **144** in yields ranging from 48% to 86% and with high stereocontrol (up to 94:6 dr, up to 99% ee).

Next, the reduction and hydrolysis of α-nitrophosphonate **144** (Ar = *p*-PhOC_6_H_4_) under acidic conditions results in the formation of enantioenriched α,β-diaminophosphonate **145** in 84% yield with retention of the absolute configuration. According to the authors, the chiral Brönsted acid catalyst activates both the imine and nitro phosphonate substrates through hydrogen bonding. Since both nitro and phosphoryl groups may be activated by acid catalysts, a bulky phosphonate is selected in order to minimize the activation of the phosphoryl group enhancing the steric repulsion. The nitro group is, in this way, located close to the catalyst and the **144**-*anti* product is favored through transition state **TS10**, while the high bulkiness of the phosphonate moiety prevents the formation of **TS11** and **144**-*syn* isomer is obtained as the minor product.

As a continuation of this work, Namboothiri and colleagues reported in 2012 the conjugate addition of α-nitrophosphonates **146** to α,β-unsaturated ketones **147** uing in this case cinchona alkaloid-derived thiourea **VI** as a bifunctional catalyst (Scheme 27) [62]. For this transformation, a tentative model of addition is proposed by the authors, where both acidic protons of the thiourea moiety activate ketones by means of hydrogen bonding, while the basic nitrogen of the quinuclidine unit stabilizes the nitrophosphonate anion (**TS12-13**). Although the reaction affords optically active α-nitrophosphonates **148** in high yields (70–97%), the enantioselectivity of the reaction is found to be strongly dependent on the substituent of the ketone. Thus, when aromatic groups bearing electron-donating substituents are used, enantioselectivities ranging from 70% to 94% are obtained. In contrast, the use of some electron poor aromatic groups, such as 4-nitrophenyl, heteroaryl substituents such as 2-furyl or 2-thienyl, and aliphatic cyclohexyl substituents results in a drastic decrease into the enantioselectivity (35–44%).

Following their interest in this reaction, a few years later, the same authors proposed that the use of stronger acidic squaramide catalyst **VII** in the reaction improves the poor enantioselectivity when electron-withdrawing groups are used [63] (Scheme 27). A relevant improvement not only on the stereocontrol but also on the reactivity was reported. Thus, yields above 90% and enantioselectivities ranging from 85% to 99% for all the aromatic and heteroaromatic ketones except for 2-substituted aryl groups such as 2-Cl-C_6_H_4_ (97%, 74% ee) and 1-naphthyl (98%, 15% ee) are obtained. In contrast, cyclohexyl substituted enone only affords moderate yield and enantioselectivity (70%, 51% ee).

In addition, the authors reported several useful transformations of the obtained optically active α-nitrophosphonates into α-aminophosphonates (Scheme 28). For instance, the reduction of the nitro group in the presence of zinc and ammonium chloride results in the unprotected amino group that spontaneously leads to the formation of cyclic imine **149** in 86% yield.

On the other hand, the Baeyer–Villiger oxidation allows obtaining of the corresponding ester **150** in almost quantitative yield. This ester **150** can be subsequently used for the synthesis of cyclic lactam **151** after reduction of the nitro group and subsequent intramolecular lactamization reaction (Scheme 28). Moreover, the reaction of ester **150** with a primary amine to yield the acyclic amide **152**, followed by the Clemmensen reduction of the nitro group affords the acyclic α-aminophosphonate derivative **153**. 

Following the same line, in 2013, Jászay and colleagues reported the addition of α-nitrophosphonates **154** to aryl acrylates **155** using also an squaramide organocatalyst **VIII** (Scheme 29) [64]. Even though some bulky phosphonates were tested in the reaction with phenyl acrylate, the use of *iso*-propyl and butyl phosphonates does not result in a further improvement on the reaction yield or enantiocontrol (82–85%, 52–64% ee) and the best enantiomeric excesses are obtained with ethyl phosphonates (93%, 76% ee).

Concerning aryl acrylate substrates, the best enantioselectivities are obtained for electron-donating aryl groups (e.g., 2,6-(OMe)_2_C_6_H_3_, 92%, 96% ee). In contrast, although no relevant effect is observed on the reaction yield, the use of strongly electron-withdrawing aromatic rings such as 2-NO_2_C_6_H_4_ results in a lower stereocontrol (90%, 40% ee). Besides, the reduction of the nitro group results in a mixture of phosphorus substituted γ-lactam **151** and cyclic α-iminophosphonate **157** in variable ratios depending on the reaction pressure and the aryl groups present on the ester moiety.

Other Michael acceptors different than conjugated carbonyl compounds have been used in enantioselective reactions with α-nitrophosphonates. Specifically, in 2012 Mukherjee’s group reported the use of thiourea–alkaloid bifunctional catalyst **IX** for the addition of α-nitrophosphonates **146** to conjugated nitroalkenes **158** to provide tetrasubstituted α-nitrophosphonates **159**, in yields ranging from 64% to 82% when using both, aryl and alkyl groups on the nitroalkene (Scheme 30) [65]. Curiously, when 2-naphthyl nitroalkene is used, the corresponding α-nitrophosphonate **159** is formed in only 38% yield. Nevertheless, diastereomeric ratios ranging from 83:17 to 95:5 and enantiomeric excesses up to >99% are obtained in all cases. The concomitant reduction of both nitro groups results in an intramolecular cyclization reaction, which leads to chiral pirazolidine **160** in 60% yield with no loss of the optical purity.

In 2013, the use of vinyl sulfones as Michael acceptors in the addition of α-nitrophosphonates **146** was simultaneously reported by Namboothiri and Lu (Scheme 31) [66,67].

Namboothiri proposes an enhancement of the electrophilicity of vinyl sulfone substrates through the establishment of two hydrogen bonds between the squaramide catalyst **X** acidic protons and the sulfone oxygen atoms, while the basic nitrogen of the alkaloid moiety activates the α-nitrophosphonate as a nucleophile (Scheme 31, **TS14**). The reaction products are obtained in this case in excellent yield and enantiocontrol when aryl sulfones **161** are used (85–99%, 90–98% ee). In contrast, the use of tetrazole-derived sulfone results in a decrease in the enantiomeric excesses (74–79% ee). Moreover, the reduction of the nitro group affords α-aminophosphonate **163** in almost quantitative yield (95%). Slightly lower yields but similar enantiomeric excesses were obtained by Lu and colleagues by using thiourea catalyst **XI** (50–98%, 86–95% ee), obtaining, in this case, the opposite enantiomer (*S*)-**162**.

Analogously to α-nitrophosphonates, α-isothiocyanatophosphonates **164** possess a significantly acidic proton in the alpha position, which makes them suitable to be used as nucleophiles. In this regard, Yuan and colleagues reported in 2013 the enantioselective addition of α-isothiocyanatophosphonates **164** to aldehydes **165** catalyzed by bifunctional thiourea catalyst **IX** (Scheme 32) [68]. In this case, the initial nucleophilic addition to aldehydes **165** proceeds in a similar manner as in the case of α-nitrophosphonates, through the activation of aldehyde electrophile**165** by the thiourea acidic protons and a simultaneous deprotonation of α-isothiocyanatophosphonates **164** by the quinuclidine basic unit of the catalyst (**TS15**). Due to the electrophilic character of the central carbon in isothiocyanates, a subsequent intramolecular addition of the alcohol occurs (**TS16**), leading to the formation of cyclic α-aminophosphonates **166** in moderate to good yield and diastereocontrol (36–93%, 84:16 to >99:1 dr). However, only moderate enantiomeric excesses are obtained (68–81% ee) if aromatic aldehydes are used. Besides, the use of acetaldehyde as electrophile results in a dramatic drop into the enantioselectivity (66%, 85:15 dr, 55% ee).

One year later, Wang and colleagues reported an improvement in Yuan’s work using in this case squaramide catalyst **XII** (Scheme 33) [69]. In this reaction, they obtain cyclic α-aminophosphonates **169** in excellent yield and stereocontrol using aldehydes **165** bearing not only electron-donating and electron-withdrawing groups, but also heteroaryl (2-furyl, 2-thienyl) aldehydes or conjugated cinnamaldehyde (86–99%, 94:6 to >95:5 dr, 87 ≥ 99% ee). Moreover, they also extended this methodology to *N*-tosyl aldimines **168**, cyclic thioureas **170** with similar results (80–99%, 86:14 to 92:8 dr, 92 ≥ 99% ee).

More recently, Albrecht and colleagues described the synthesis of tetrasubstituted spirocyclic chiral α-aminoesters and α-aminophosphonates **173** through a conjugate addition of α-isocyanates **171** to conjugated barbiturates **172** in the presence of squaramide catalyst **X** (Scheme 34) [70]. Although the scope is limited, spirocyclic α-aminophosphonates **173** are obtained in high yields and stereocontrol (60–99%, 88:12 to >95:5 dr, 92–98% ee).

So far, the reactions described in this chapter, giving an account of reactions where the key step implies the formation of a C-C bond, entail the addition of an α-aminophosphonate equivalent onto an electrophile (Figure 4a; vide supra). A complementary general method involving the generation of C-C bonds that also leads to tetrasubstituted α-aminophosphonates consists of the addition of carbon nucleophiles to α-iminophosphonates (Figure 4b; vide supra) [71].

The synthesis of activated α-ketiminophosphonate substrates is known to be a challenging task, mainly due to the low reactivity found in the typical amine-carbonyl condensation reactions, where deactivated amide substrates are required, and the intrinsic tendency of α-ketophosphonates to eliminate the phosphonate substituent, which leads to acylation reactions [72]. Moreover, the high moisture sensitivity of such substrates entails additional obstacles for the purification of the imines which very often have to be prepared in situ. For this reason, it was not until 2012 when our research group reported an efficient synthesis of α-ketiminophosphonates **174** and the enantioselective addition of nucleophiles to such substrates (Scheme 35) [73]. In this reaction, the cinchonidine (**XIII**)-catalyzed nucleophilic addition of cyanide to α-phosphorated ketimines **174** provides optically active α-cyano α-aminophosphonates **176** in high yield (75–80%) and enantioselectivities ranging from 73% to 92%. The presence of bulky isopropyl groups was found to be crucial in order to obtain high enantiocontrol if compared with other alkyl and aryl phosphonates. The fact that in alcoholic solvents, the reaction proceeds fast but with no enantiocontrol might indicate a crucial role for the hydroxyl group of cinchona alkaloid in the transition state **TS17**, which may activate the substrate via hydrogen bonding with the iminic nitrogen. In addition, optically active α-aminophosphonic acid **177** was obtained in 80% yield without racemization by the hydrolysis reaction of the cyano group under strong acidic conditions.

As a continuation of our research on the enantioselective nucleophilic addition reactions to α-ketiminophosphonates, a few years later, we reported the asymmetric aza-Henry reaction with ketimines **178** (Scheme 36) [74,75,76], using bifunctional thiourea-alkaloid a catalyst **XIV**. The reaction allows the use of electron-donating and electron-withdrawing aromatic groups at the imine substrates with no relevant differences in the yield or the enantioselectivity of the obtained α-amino-β-nitrophosphonates **180** (82–87%, 80–84% ee). In addition, the reduction of the nitro group is also reported, leading to α,β-diaminophosphonate **181** in almost quantitative yield.

In the same context, more recently, we described the first example of an enantioselective aza-Reformatsky reaction with non-cyclic ketimines **178**, using dialkyl zinc reagents and BINOL-derived chiral ligand **XV**. The presence of molecular oxygen is crucial in this case in order to obtain a high yield, since other byproducts are observed when an inert atmosphere is used. The reaction can be successfully generalized to several aryl and heteroaryl ketimines **178** and alkyl iodoacetates **182**, affording tetrasubstituted α-aminophosphonates **183** in excellent yield and enantiocontrol (76–92%, 93–>99% ee). Furthermore, the synthesis of β-lactam **185** containing a tetrasubstituted α-aminophosphonate is also described by a selective deprotection of the ester group and subsequent lactamization reaction.

Although enantioselective nucleophilic additions to α-alkyl iminophosphonates remains almost unexplored, in 2014, a particular case using α-trifluoromethyl α-iminophosphonates was reported by Onys’ko and colleagues (Scheme 37) [77]. In particular, proline (**I**)-catalyzed nucleophilic addition of acetone (**33**) to *N*-unprotected α-iminophosphonate **186** yields tetrasubstituted α-iminophosphonate **187** in high yield and enantiocontrol (80%, 90% ee). Moreover, some further transformations of α-aminophosphonate **187** are reported by the authors. For instance, the reaction of substrate **187** with 2,5-dimethoxyfuran in acid media leads to *N*-heterocyclic derivative **189** in 84% yield through an aldol reaction of the in situ generated pyrrole **188**. On the other hand, the reaction with aryl isocyanates leads to pyrimidine **191** via urea intermediate **190**. As in the previous case, an intramolecular condensation involving the ketone moiety affords substrate **191** in 89% yield without racemization.

Following a similar approach, Ohshima and colleagues reported some examples of a rhodium complex **XVI**-catalyzed enantioselective alkynylation of α-CF_3_ α-iminophosphonates **192** (Scheme 38) [78]. Once the alkyne **193** is inserted into the catalyst by displacement of TMS substituted alkyne (**195**), an enantioselective alkynylation of imine **192** takes place leading to amide–rhodium complex **196**, where a new chiral center is formed. Then, a new insertion occurs by the introduction of a second unit of alkyne **193**, leading to the formation of rhodium complex **197**. Finally, amide deprotonation of the terminal alkyne ends with the formation of α-alkynyl α-iminophosphonates **194** and the consequent regeneration of the active catalyst **195**. The use of aryl and cyclopropyl provides α-aminophosphonates **194** in excellent yields and enantioselectivity (86–99%, 80–93% ee).

Following a similar approach, in 2012, Che’s group reported the use of chiral rhodium catalysts **XVII** and **XVIII** on the enantioselective multicomponent reaction of diazophosphonates **198**, anilines **199** and aromatic aldehydes **165** (Scheme 39) [79]. Here, in the first reaction step, the rhodium catalyst reacts with diazo compound **198** to form rhodium carbene species **201** with the release of nitrogen gas. The subsequent insertion of aniline moiety leads to an ionic intermediate **202**, which easily evolves to zwiterionic species **203**. At this point, the rhodium-phosphonate undergoes an addition reaction to the corresponding aldehyde substrate **165**, leading to optically active tetrasubstituted α-amino-β-hydroxy phosphonates **200** while the catalyst unit is released for a new catalytic cycle. In this reaction, tetrasubstituted α-aminophosphonates **200** are obtained in moderate to excellent yield and stereocontrol (56–86%, 76:24 to 94:6 dr, 60–98% ee).

During the last lustrum, a new family of cyclic α-ketiminophosphonates **204** has been used as electrophilic sources in enantioselective nucleophilic addition reactions. In particular, the palladium–**XIX**-catalyzed enantioselective arylation reaction of α-iminophosphonates **204** was reported in 2016 by Zhou’s group (Scheme 40) [80].

The reaction can be generalized to several α-iminophosphonate substrates **204** and boronic acids **205**, bearing electron-donating and electron-withdrawing aryl groups. In consistence with other reported examples, the use of bulky *iso*-propyl phosphonates results in higher enantiocontrol, providing cyclic α-aminophosphonates **206** in excellent yields (73–97%) and enantioselectivities above 99%. The high steric bulkiness of the phosphonate moiety induces the coordination of the catalyst to the imine group through *Si*-face (**207**) since the coordination through *Re*-face (**208**) implies steric repulsions not only between phosphine and phosphonate moieties but also between *tert*-butyl group at the oxazoline ring and sulfonyl protecting group at the imine group.

One year later, the enantioselective Friedel–Crafts reaction of indoles **210** to five-membered imines **209** was also reported (Scheme 41) [81]. In this case, phosphoric acid **XX** was selected as the optimal catalyst, affording optically active α-aminophosphonate functionalized indoles **211**. The reaction can be successfully generalized to several indole substrates bearing electron-donating and electron-withdrawing groups with excellent yields and enantiocontrol (85–98%, 87–98% ee). However, the use of 2-methylindole results in a drastic drop in the enantioselectivity (91%, 59% ee). In addition, the addition of simple pyrrole to imines **209**, leads to the formation of the analogous α-aminophosphonate substituted pyrroles in 98% yield with 84% enantiomeric excess. 

In the same context, Zhang and colleagues described in 2018 a single example on the enantioselective Mannich-type addition of glycine Schiff bases **213** to five-membered iminophosphonates **212** (Scheme 42), providing tetrasubstituted α-aminophosphonate **214** in moderate yield and stereocontrol (48%, 80:20 dr, 83% ee) [82].

In the same year, Ma and colleagues reported the enantioselective decarboxylative addition of β-keto acids **215** to cyclic α-iminophosphonates **204** (Scheme 43) [83]. The reaction affords α-amino-β-ketophosphonates **216** when five- or six-membered cyclic imines **204** are used as electrophile substrates. In addition, several alkyl and (hetero)aryl keto acids were tested, obtaining in all cases excellent yields and enantiocontrol (77–93%, 90–99% ee), and allowing a decrease in the catalyst loading down to 1% without any lose in the enantioselectivity. Likewise, the reduction of the ketone to obtain chiral alcohol **217** (88%, 94:6 dr) and the synthesis of aziridine **218** in presence of *tert*-butyl hydroperoxide (61%, single diastereoisomer) are described with high yields and diastereocontrol. 

Besides reactions that imply C-C bond formation, through the functionalization of trisubstituted α-aminophosphonate derivatives with electrophiles or by the addition of nucleophiles to α-iminophosphonates, cycloaddition reactions are also efficient protocols leading to the formation of optically active tetrasubstituted α-aminophosphonate derivatives. The first example of such reaction was published in 2011 by Kobayashi (Scheme 44) [84]. In particular, they reported a [3+2] reaction between Schiff bases **219** and *tert*-butyl acrylate **220**, using silver hexamethyldisilazane as a catalyst and chiral bisphosphine ligand **XXIII**. The reaction can be successfully generalized to several chiral pyrrolidines **221** showing a tetrasubstituted α-aminophosphonate moiety in excellent yields and stereocontrol (72–81%, >99:<1 dr, 90–98% ee). According to the authors, the in situ formed silver–**XXIII** complex catalyzes the enolization of the phosphoryl group, leading to the active reagent for the *exo*-[3+2] cyclization through transition state **TS18** (Scheme 44).

More recently, an example of a dipolar cycloaddition was reported by Peng and colleagues (Scheme 45) [85]. The reaction between diazophosphonates **222** and acryloyl oxazolidones **223** in presence of Mg–**XXIV** complex afforded enantioenriched pyrazolines **224** in high yield and enantiocontrol.

For this reaction, the authors propose an activation of the electrophile through a double coordination of the chiral Mg-complex to both carbonyl groups of acryloyl oxazolidones **223**, inducing the preliminary addition of the nucleophile from the *Re*-face in **TS19**. The subsequent trapping of the in situ formed enolate by the nitrogen atom of the diazo compound leads to cyclic α-aminophosphonates **224** in moderate to excellent yield and enantiocontrol (52–93%, 74–95% ee).

In addition, the double reduction of the pyrazoline and oxazolidone moieties, followed by treatment with carbonyl di-imidazole (CDI), leads to bicyclic pyrazolidine **225** in 84% yield. On the other hand, Boc-protected pyrazoline **226** can be also obtained in high yield (86%) by treatment with *terc*-butyl dicarbonate. Next, the selective deprotection of oxazolidone moiety by a reduction reaction with sodium borohydride, and the subsequent protection of the resulting alcohol affords pyrazoline **227** in 63% yield. Finally, the reduction of pyrazoline, and the in situ Cbz-protection of the newly formed NH group leads to pyrazolidine **228** in 86% yield.

Moreover, although not strictly a cycloaddition reaction, in 2018, Chi’s group reported the use of five- and six-membered cyclic α-iminophosphonates **229** in a *N*-heterocyclic carbene **XXV**-catalyzed formal [4+2] cycloaddition with α,β-unsaturated aldehydes **230** (Scheme 46) [86,87].

In the catalytic cycle proposed by the authors, after the activation of carbene catalyst **XXV***, the reaction starts with an addition of the carbine to aldehyde substrate to generate intermediate **233**. Then, a stoichiometric amount of an oxidant reagent is needed to re-generate the carbonyl group in species **234**. Next, a basic source generates enolate **235** and the enantioselective vinylogous addition to the imine takes place giving rise to adduct **236**. The cycle ends with an intramolecular addition of the nitrogen atom to the carbonyl moiety, which leads to cyclic α-aminophosphonates **231** or **232**, after releasing the active carbene catalyst **XXV***. The reaction products were obtained in moderate to excellent yield (51–96%) and enantioselectivities above 92%. Remarkably, simultaneously to Chi’s work, Ye and colleagues reported the same transformation in similar reaction conditions with a different NHC catalyst, resulting in the formation of the opposite enantiomer of α-aminophosphonates **231** and **232** [88].

### 3.2. C-P Bond Formation

One of the simplest strategies for the preparation of α-aminophosphonates is the addition of phosphorus nucleophiles to imines. Although the first catalytic asymmetric hydrophosphonylation of aldimines was described by Shibasaki in 1995 [89], it is easy to understand the further complication of developing a catalytic system that works on ketimines, due to the more difficult discrimination between both faces in the prochiral species if compared to aldimines [90,91,92]. Therefore, it was not until 2009 when Nakamura’s group described the first nucleophilic addition of diphenyl phosphite to *N*-sulfonyl ketimines **237** catalyzed by cinchona alkaloids **XXVI** and **XXVII** (Scheme 47) [93]. In the presence of a base and 2% of alkaloid **XXVI,** tetrasubstituted α-aminophosphonates **238** in excellent yields (93–99%) are obtained, with moderate to excellent enantiomeric excesses (55–97% ee). The use of hydroquinidine **XXVII** epimer as organocatalyst results equally in the formation of α-aminophosphonates **238** with the opposite configuration, in yields up to 99% with good stereocontrol (52–95% ee). Since it was evidenced that the reaction does not work without the presence of a base, the transition state **TS20** proposed by the authors might consist in a coordination between the alkaloid nitrogen with the sodium cation that would improve the nucleophilic character of the phosphite reagent. Furthermore, the use of an alkaloid with a protected alcohol group results in a drop in the enantioselectivity, which may indicate a dual activation mode of the alkaloid in which the hydroxyl group activates the imine **237** by hydrogen bonding.

A few years later, Shibasaki reported the hydrophosphorylation reaction of *N*-thiophosphinyl imines **239** with various phosphites, using copper complexes and chiral bis-phosphine-based ligands as catalysts, thus obtaining tetrasubstituted α-aminophosphonates **240** with excellent enantioselectivities ranging between 86% and 97% (Scheme 48) [94,95]. The reaction works very efficiently even using only 0.5–2% copper–**XXVIII** catalyst, which can also be reused.

Another approach that can be used for the hydrophosphorylation reaction of α-phosphorated ketimines consists in the use of *N*-phosphinyl imines **241** in the presence of a bifunctional thiourea–iminophosphorane catalyst **XXIX** (Scheme 49) [96]. This type of catalyst has a superbase iminophosphorane acceptor and a classical thiourea donor unit. As described by the authors, the proposed transition state **TS21** may involve the initial deprotonation of the phosphite reagent by the superbase unit, while the thiourea donor activates the imine electrophile **226** by hydrogen bonding. The resulting tetrasubstituted α-aminophosphonates **242** are obtained in excellent yields (78–99%) and with moderate to good enantiomeric excesses (46–71% ee).

The first example of a hydrophosphonylation reaction of cyclic ketimines was described in 2013 [97]. In this case, trifluoromethylimines **243** derived from quinazolinone react with methyl, ethyl, benzyl or phenyl phosphites in the presence of a bifunctional alkaloid-thiourea catalyst **IX** (Scheme 50). The reaction is highly dependent on the medium and chloroform, dichloromethane or hexane: dichloromethane (5:1) mixtures were used depending on the phosphite reagent. In this reaction, α-aminophosphonates **244** were obtained in high yields (75–91%) and excellent stereocontrol (81–93% ee). In the proposed transition state **TS22**, the two hydrogens are in a *gauche* conformation with respect to the bulkiest groups, in order to minimize steric interactions. The basic amine and the donor thiourea adopt also a *gauche* conformation, so that they can simultaneously activate the nucleophilic phosphite and the electrophilic imine **243**, through an acid–base interaction and a double hydrogen bond of the thiourea moiety with nitrogen and carbonyl, respectively. This rigid conformation directs the nucleophilic attack from the *Re-*face and supports the stereochemical outcome of the addition adduct with the *R* configuration.

Later on, the addition of diphenyl phosphite to isatin-derived ketimines **245** using a bifunctional organocatalyst derived from squaramide **XXX** was described with high yields (85–96%) and excellent stereocontrol (83–97% ee) (Scheme 51) [98].

The use of 4-Br or 5-NO_2_ as substituents into the phenyl ring of the isatin derivatives decreases notably the enantioselectivities in this reaction, with excesses between 52% and 68%. Taking into account the poor results obtained for the addition of phosphites to the analogous ketones, the authors figure out an essential role of the iminic protecting group and propose transition state **TS23**, where the squaramide unit binds with isatin-derived imine while quinuclidine moiety deprotonates the phosphite reagent, in order to facilitate the reaction. A preferential *Re-*face attack of the phosphite to ketimine **245** affords the (*R*)-isomer of α-aminophosphonates **246**.

In the same year, a similar approach was described by Chimni’s group, that is, the addition of diphenyl phosphite to ketimines **245**, mediated by *Chinchona*-derived catalyst **XXXI,** to provide similar yields (72–88%) and enantiomeric excesses (71–97% ee) (Scheme 51) [99].

Another example regarding the addition of phosphites to isatin-derived imines **245** was described by Kim in 2016, using a bifunctional squaramide **XXXII** originated from binaphthyl (Scheme 51) [100]. The catalytic system is very efficient for unsubstituted isatins or those substituted with alkyl groups, obtaining yields up to 94% and enantioselectivities ranging from 73% to 99%. It should be noted that the use of an allyl or benzyl protecting group at the nitrogen of the amide moiety is strongly relevant for this process, since the presence of a carbamate group (R^1^ = Boc) in that position results in a strong drop in the yield (45%) and almost a total loss of stereocontrol (26% ee). Following the line of the previous examples, the proposed transition state is expected to be similar to **TS23**, showing a bifunctional activation of the phosphite and the ketimine **245**. In this conformation, the phosphite nucleophile may attack preferentially the *Re-*face of the imine.

Some enantioselective methods for the addition of phosphites to imines derived from isatins **245**, making use of metallic catalysts instead of organocatalysis have been also developed. In 2016, titanium–Salen complex Ti-**XXXIII** was used as catalyst, in the hydrophosphorylation reaction of ketimines **245** to provide tetrasubstituted α-aminophosphonates **246** with high yields (84–88%) and good enantiomeric excesses (63–99%) (Scheme 51) [101]. It should be noted that the amide group of the isatin moiety must be substituted, since the presence of an unprotected nitrogen resulted in a significant decrease in enantiomeric excess (46% ee), although without affecting the yield (88%). 

A useful strategy for the in situ generation of cyclic ketimines and their subsequent activation for the nucleophilin addition of a phosphite reagent was reported by Singh in 2017 (Scheme 52) [102]. In this report, ketimines are produced from α-hydroxyamines **247** and then activated by the presence of phosphoric acid catalyst **XXXIV** as in **TS24**, leading to the formation of tetrasubstituted isoindolinones **248**, with yields up to 98% and high enantioselectivities (72–97%).

Furthermore, the addition of diphenyl phosphite to azirines **249** was described a few years ago, using zinc complexes with chiral bisimidazolines **XXXV**, with yields up to 99% and enantiomeric excesses between 80% and 96% (Scheme 53) [103]. The resulting phosphorus-substituted aziridines **250** were converted to oxazolines **253** through an initial acylation of the nitrogen with 3,5-dinitrobenzoyl chloride (**251**), followed by treatment with boron trifluoride. Additionally, the ring opening of aziridines **250** with hydrobromic acid under ultrasound treatment, leads to the corresponding β-bromo α-aminophosphonates **254**. The halogen atom can be also removed by a radical reaction to give α-aminophosphonate **255** without any loss of enantiomeric purity (92% ee).

The proposed pathway for this transformation may consist of an initial formation of a Zn(II)–**XXXV** complex **256** in which the metal coordinates with the oxygen and only one of the imidazoline moieties of the ligand, leaving the other nitrogen atom free. The zinc atom then coordinates with azirine **249** in a tetrahedral mode (**TS25**), and the phosphite moiety establishes a hydrogen bond with the second free imidazoline unit in **TS26**. Azirine is therefore positioned with its substituent facing outwards, favoring a *Re-*face attack of the nucleophile. After the addition and formation of the corresponding phosphorus aziridines **250** the Zn catalyst is released, in order to be able to return to the catalytic cycle (Scheme 54).

### 3.3. C-N Bond Formation

Although there are only a few reports in the literature, the electrophilic amination reaction of trisubstituted phosphonates can also be an effective method for the preparation of tetrasubstituted α-aminophosphonates.

In 2005, Jørgensen and Kim described, almost simultaneously, the electrophilic amination of β-ketophophosphonates with azodicarboxylates. Jørgensen’s method makes use of a zinc–oxazolidine complex **XXXVI,** in order to generate an enolate from β-ketophosphonate **257**, which undergoes an enantioselective nucleophilic addition to the nitrogen electrophile **258**, providing the corresponding α-aminophosphonates **259** with yields up to 98% and excellent stereocontrol (85–98% ee) (Scheme 55) [104].

On the other hand, Kim described a similar reaction using a chiral palladium-complex **XXXVII** in order to catalyze the addition of β-ketophosphonates **260** to diethyl azocarboxylates **261** with similar yields (68–92%) and enantiomeric excesses (99% ee) (Scheme 56) [105].

Finally, although only one example is reported with an acceptable yield (72%) and moderate enantioselectivity (50% ee), phosphonate-substituted aziridine **265** is obtained from α, β-unsaturated β-ketophosphonates **263** using a bifunctional catalyst derived from thiourea **XXXVIII** (Scheme 57) [106]. The transition state **TS27** might consist of a double activation of the ketophosphonate **263** by the thiourea unit while the basic unit deprotonates the nitrogen of the hydroxylamine derivative **264**. Then, the amphiphilic oxycarbamate reacts with the olefin through a conjugate addition reaction, while the enolate attacks the nitrogen atom of the amine, with concomitant release of the tosyl group.

## 4. Final Remarks

Even though some examples of stereocontrolled synthesis of tetrasubstituted α-aminophosphonates have been reported, they are still rather limited if compared with the homologous reactions for the preparation of trisubstituted α-aminophosphonates. In particular, in recent years, the main efforts have been focused on the enantioselective transformations, which are known to be more attractive than diastereoselective ones. It should be noted that most of the enantioselective reactions summarized in this review were published during the past decade and thus, more articles regarding this approach are expected in the following years.

One of the most promising topics is related to nucleophilic additions to α-phosphorylated ketimines, which experienced an important growth during the last lustrum. Another promising topic is the enantioselective addition of phosphorus nucleophiles to ketimines. It has been slightly explored, with just a few examples reported to date, but due to the vast number of synthetic protocols for the preparation of imines known in the literature, the development of new enantioselective protocols for this transformation would constitute a relevant improvement in order to expand the structural diversity of tetrasubstituted α-aminophosphonates.

## Data Availability

Not applicable.

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
