# Peer review of "Asymmetric Synthesis of Tetrasubstituted α-Aminophosphonic Acid Derivatives"

_molecules, 2021, doi:10.3390/molecules26113202_

Round 1
Reviewer 1 Report
The manuscript submitted by Vicario and coworkers overviews recent advances in the asymmetric synthesis of quaternary alpha-aminophosphoric acids and their derivatives as useful building blocks and biologically active molecules. The reported works are organized in a concise way and summarized in the scrupulous reference section, while the classification approach described in L. 61–70 (and Figure 4) is similar to that in a standout review for the closely related topics. The authors need to introduce the following review articles in context of the scope of this manuscript in L. 52: Bera, K.; Namboothiri, I. N. N. Asian J. Org. Chem. 2014, 3, 1234–1260; Ordóñez, M.; Sayago, F. J.; Cativiela, C. Tetrahedron 2012, 68, 6369–6412.
Other than this point, I believe that this review will enhance the knowledge in the field and attracts considerable attention from the readers. Overall, the material merits publication. after addressing the following issues.
1) Most of equations and schemes should be revised for publication. For example, overlapped texts in Schemes and bond drawings of P=O groups are not qualifiable.
2) L. 183, 272, 299, 412; unify the nomenclature of compound 40: Replace “methylbenzylamine” and “phenylethylamine” with “1-phenylethylamine”,
3) There are various typos and grammatical errors. A careful check by an expert proof reader should be completed. Several errors are listed below:
L. 29; remove a space from “a-amino phosphonic”.
L. 49; replace “Most of the existing literature to date in this field is …” with “Most of the existing literatures to date in this field are …”.
L. 85; change “piruvaldehyde” with “pivalaldehyde”.
L. 97; replace “reduction the amide carbonyl” with “reduction of the amide carbonyl”.
L. 112; change “mayor” with “major”.
L. 123; change “2,5 dimethoxy-tetrahydrofuran” with “2,5-dimethoxytetrahydrofuran”.
L. 125; change “furane” with “furan”.
L. 129; change “an alkyl halide” with “alkyl halides”.
L. 149; change “afords” with “affords”.
In Scheme 4; change “2-naphtyl” with “2-naphthyl”.
L. 174; change “indol” with “indole”.
L. 182; change “imines 41” with “imine 41”.
L. 183; change “triphenyl phosphine” with “triphenylphosphine”.
L. 185, 286–287, 300, 311–312; change “triethylphosphite” with “triethyl phosphite”.
L. 203; change “diastereoselectivityin” with “diastereoselectivity in”.
L. 212; change “diethylphosphite” with “diethyl phosphite”.
L. 222; change “tert-butyl sulfinimines 51” with “tert-butylsulfinyl imines 51”.
L. 235; change “potassium amidure (KHMDS)” with “potassium bis(trimethylsilyl)amide (KHMDS)”.
L. 249; change “diphenylphosphite” with “diphenyl phosphite”.
L. 342; change “dioxaphospholanedicarboxilate 92” with “dioxaphospholanedicarboxylate 92”.
L. 353; change “sec-butyl lithium” with “sec-butyllithium”.
L. 379; remove the first name, “Maurice”.
L. 389; change “acrylaldehyde” with “acrolein”.
L. 429; change “n-butyl ammonium” with “tetra-n-butylammonium”.
L. 431; change “tert-butanol” with “tert-butyl alcohol”.
L. 438, 441, 443–444, 447, 451–452, 459; change “nitro phosphonoate” with “nitro phosphonoate”.
L. 497; change “in mixtures of” with “in a mixture of”.
L. 585; change “amino phosphonoate” with “aminophosphonoate”.
L. 602; change “aminophosphonoates 187” with “aminophosphonoate 187”.
L. 626; change “carbine” with “carbene”.
L. 627; change “ionic intermediate” with “an ionic intermediate”.
L. 635; change “a family of have” with “has”.
L. 663; “alpha-Aminophosphonate 214”.
L. 688; “alpha-aminophosphonate 221”.
L. 689; lowercase “silver-XXIII”.
L. 741; change “alcaloid” with “alkaloid”.
L. 773; change “trifliuoromethyl imines” with “trifliuoromethylimines”.
L. 799–800; remove “preferentially.
L. 845; put a period at the end of sentence.
L. 860; remove “a”.
L. 865; replace the caption with “Enantioselective amination of b-ketophosphonates 257 catalyzed by zinc-oxazolidine complex.”.
Author Response
Following reviewer 1’s suggestion, we have added the following references (references 23 and 24): Bera, K.; Namboothiri, I. N. N. Asian J. Org. Chem. 2014, 3, 1234–1260; Ordóñez, M.; Sayago, F. J.; Cativiela, C. Tetrahedron 2012, 68, 6369–6412.
1) Most of equations and schemes should be revised for publication. For example, overlapped texts in Schemes and bond drawings of P=O groups are not qualifiable.
All the schemes have been revised in order to clarify the P=O groups. Regarding the overlapped texts, It seem to be an error when converting to the pdf file, but no overlapped text were found in the original word document.
2) L. 183, 272, 299, 412; unify the nomenclature of compound 40: Replace “methylbenzylamine” and “phenylethylamine” with “1-phenylethylamine”,
The nomenclature for compound 40 has been unified as “1-phenylethylamine”.
3) There are various typos and grammatical errors. A careful check by an expert proof reader should be completed. Several errors are listed below:
All the reviewer’s comments have been checked and corrected. A detailed list of the changes can be checked bellow. Other additional errors have been also corrected.
- 29; remove a space from “a-amino phosphonic”.
- Corrected
- 49; replace “Most of the existing literature to date in this field is …” with “Most of the existing literatures to date in this field are …”.
- Corrected
- 85; change “piruvaldehyde” with “pivalaldehyde”.
- Corrected
- 97; replace “reduction the amide carbonyl” with “reduction of the amide carbonyl”.
- Corrected
- 112; change “mayor” with “major”.
- Corrected
- 123; change “2,5 dimethoxy-tetrahydrofuran” with “2,5-dimethoxytetrahydrofuran”.
- Corrected
- 125; change “furane” with “furan”.
- Corrected
- 129; change “an alkyl halide” with “alkyl halides”.
- Corrected
- 149; change “afords” with “affords”.
- Corrected
- In Scheme 4; change “2-naphtyl” with “2-naphthyl”.
- Corrected
- 174; change “indol” with “indole”.
- Corrected
- 182; change “imines 41” with “imine 41”.
- Corrected
- 183; change “triphenyl phosphine” with “triphenylphosphine”.
- Corrected
- 185, 286–287, 300, 311–312; change “triethylphosphite” with “triethyl phosphite”.
- Corrected
- 203; change “diastereoselectivityin” with “diastereoselectivity in”.
- Corrected
- 212; change “diethylphosphite” with “diethyl phosphite”.
- Corrected
- 222; change “tert-butyl sulfinimines 51” with “tert-butylsulfinyl imines 51”.
- Corrected
- 235; change “potassium amidure (KHMDS)” with “potassium bis(trimethylsilyl)amide (KHMDS)”.
- Corrected
- 249; change “diphenylphosphite” with “diphenyl phosphite”.
- Corrected
- 342; change “dioxaphospholanedicarboxilate 92” with “dioxaphospholanedicarboxylate 92”.
- Corrected
- 353; change “sec-butyl lithium” with “sec-butyllithium”.
- Corrected
- 379; remove the first name, “Maurice”.
- Corrected
- 389; change “acrylaldehyde” with “acrolein”.
- Corrected
- 429; change “n-butyl ammonium” with “tetra-n-butylammonium”.
- Corrected
- 431; change “tert-butanol” with “tert-butyl alcohol”.
- Corrected
- 438, 441, 443–444, 447, 451–452, 459; change “nitro phosphonoate” with “nitro phosphonoate”.
- Reviewer’s comment is not clear and most probably he/she had a mistake while tipping phosphonate. Based in similar comments of the reviewer, we have changed “nitro phosphonate” by “nitrophosphonate”.
- 497; change “in mixtures of” with “in a mixture of”.
- Corrected
- 585; change “amino phosphonoate” with “aminophosphonoate”.
- Most probably he/she had a mistake while tipping phosphonate. Based in similar comments of the reviewer, we have changed “amino phosphonate” with “aminophosphonate”.
- 602; change “aminophosphonoates 187” with “aminophosphonoate 187”.
- Most probably he/she had a mistake while tipping phosphonate. We have changed “aminophosphonates 187” with “aminophosphonate 187”
- 626; change “carbine” with “carbene”.
- Corrected
- 627; change “ionic intermediate” with “an ionic intermediate”.
- Corrected
- 635; change “a family of have” with “has”.
- Corrected
- 663; “alpha-Aminophosphonate 214”.
- Corrected
- 688; “alpha-aminophosphonate 221”.
- Corrected
- 689; lowercase “silver-XXIII”.
- corrected
- 741; change “alcaloid” with “alkaloid”.
- Corrected
- 773; change “trifliuoromethyl imines” with “trifliuoromethylimines”.
- Most probably he/she had a mistake while tipping trifuloromethyl. We have changed ”trifluoromethyl imines” with ”trifluoromethylimines”.
- 799–800; remove “preferentially.
- Corrected
- 845; put a period at the end of sentence.
- Corrected
- 860; remove “a”.
- Corrected
- 865; replace the caption with “Enantioselective amination of b-ketophosphonates 257catalyzed by zinc-oxazolidine complex.”.
- Corrected
Reviewer 2 Report
In this review authors provide o complete review regarding the synthesis of tetrasubstituted ∝-aminophosphonic acid derivatives in asymmetric approach. This type of compounds due to their similarity to ∝-amino acids and biological activity are in the interest of many organic chemists.
The work is written legibly and with a clear ideological outline. I recommend publishing of this revision in Molecules after minor revision – in a few cases, the reader may lose the meaning of the presented statement due to linguistic deficiencies. I believe that the work should be corrected by a native speaker before publication.
Author Response
Following reviewer 2’s suggestion, all the manuscript has been scrupulously revised by English qualified researchers in order to optimize the language. The manuscript has been substantially modified and a “tracked version” of the manuscript is submitted.
Reviewer 3 Report
This is an interesting and relevant topic about the asymmetric synthesis of α-aminophosphonic acids and it seems that the compilation of experimental approaches is sound. In general, the relevant information is given. This referee is of the opinion that the manuscript is accepted in its current form.
Author Response
All the manuscript has been scrupulously revised by English qualified researchers in order to optimize the language.